# Toxicological Assessment of Roasted Coffee Silver Skin (Testa of *Coffea* sp.) as Novel Food Ingredient

**DOI:** 10.3390/molecules27206839

**Published:** 2022-10-12

**Authors:** Liane Lorbeer, Steffen Schwarz, Heike Franke, Dirk W. Lachenmeier

**Affiliations:** 1Postgraduate Study of Toxicology and Environmental Protection, Rudolf-Boehm-Institut für Pharmakologie und Toxikologie, Universität Leipzig, Härtelstraße 16-18, 04107 Leipzig, Germany; 2Chemisches und Veterinäruntersuchungsamt (CVUA) Karlsruhe, Weissenburger Strasse 3, 76187 Karlsruhe, Germany; 3Allessa GmbH, Alt-Fechenheim 34, 60386 Frankfurt am Main, Germany; 4Coffee Consulate, Hans-Thoma-Strasse 20, 68163 Mannheim, Germany

**Keywords:** coffee silver skin, novel food, coffee by-products, *Coffea arabica*, *Coffea canephora*, partial flour replacement, risk assessment, caffeine, acrylamide, toxicological review

## Abstract

Roasted coffee silver skin is a coffee by-product, the uses of which are currently limited, e.g., as fertilizer, for energy production, or animal feed. Due to a low content of fat and carbohydrates combined with a high content of fiber, polyphenols and proteins, roasted silver skin is a valuable possible food ingredient. Potential applications include partial flour replacement in bakery products, as antioxidant and providing protein or fiber sources in sports or functional foods. As no relevant consumption of isolated silver skin occurred before 1997 in the European Union (EU), it was classified as a novel food in need of premarketing approval. Novel food applications must meet legal requirements for compositional and toxicological information. This review presents information on silver skin composition and toxicological studies. Several in vitro studies and subchronic in vivo studies are available with negative results, not suggesting a need for further studies on carcinogenic effects, reproduction, or chronic toxicity. All available studies so far concluded that no toxic effects of silver skin were found or are to be expected. For a novel food application in the EU, further in vitro studies on mutagenic potential may be needed to close a formal data gap.

## 1. Introduction

Roasted coffee silver skin is a high volume by-product of the coffee industry, which may also be used as a valuable food ingredient. The search for alternative food sources has recently gained momentum due to the political situation in Europe, which has shortened the availability of wheat on the world market. The global food security crisis is particularly affecting developing countries [1,2]. Therefore, it is necessary to develop new sources of foods that are not impacted by the Ukrainian crisis. One way to do so is through the usage of by-products, which are currently mostly thrown away. The coffee by-product silver skin is one of them.

Coffee is the roasted seed of mainly two coffee species, *Coffea arabica* (arabica coffee) and *Coffea canephora* (the so-called robusta coffee), which have a fraction of approximately 70% or 30% in typical German coffee blend [3], while international trade volumes are approximately 63% for Arabica and 37% for Canephora [4]. Other coffee species have a lower economic importance and are found in small quantities on the world market. While originally based on the African continent, the plants are now cultivated worldwide between 25° north and 25° south [5].

Coffee is among the most traded commodities in the world, around 10,178 tons of coffee beans were exported in total 2020 and almost 10,000 tons were consumed in 2020/2021. The main exporting countries in the world were Brazil, Vietnam and Columbia, Africa despite being the origin of coffee plays a minor role. The main importing regions were USA and Europe [6]. In the process of producing coffee, several by-products accumulate [7]. Among them, for example, the leaves, the cherry pulp, and the wood of the coffee plant. Furthermore, more than 50% of coffee cherries are lost as a by-product during the production of coffee beans.

After harvesting the coffee cherries they can be processed dry or wet. For dry processing the cherries are dried in the sun, the coffee beans are separated from the by-products and exported. For wet processing the beans are separated from the cherry pulp in water and afterwards fermented to remove the mucilage and exported [5].

The beans are transported and the roasting occurs. During roasting, the major part of silver skin gets removed, 2% weight of the coffee bean consist of remaining silver skin after roasting (Figure 1). This is the skin fraction that lies between the two halves of the bean [8].

There are approaches to make use of these by-products, but legal regulations to bring foods onto the European market must be considered. The coffee leaves and coffee cherry pulp (or cascara) can be used for various purposes [9] and were approved as traditional foods from third countries [10,11]. The cascara was used in several forms of spiced or unspiced, aqueous or milk-based infusions in Yemen, where the infusion is called Qishr, but also in Ethiopia and Bolivia. Infusions from coffee leaves were traditionally consumed in Africa, Asia and North America. The main consuming countries are Ethiopia, Liberia, South Sudan, Indonesia and Jamaica without giving the food a special name. The spent coffee ground was considered by the responsible authority to be not a novel food [12].

Silver skin is of special interest, as it is the only by-product formed during roasting in industrialized consuming countries in considerable amounts. This makes it more easily accessible for valorization than the other by-products of coffee. There are also already established processes for the efficient removal and collection of silver skin in coffee roasteries. Figure 2 shows separated silver skin after roasting process. Therefore, the number of publications that evaluate the possible applications of silver skin has increased recently [13].

Non food applications are in cosmetics [14], as biomass [15], for making hydrogen [16] or in biopolymers [17], historically it has been used in animal feed for a long time [18]. Due to the high content of protein and fiber, silver skin is a food component of great interest [3]. The physical properties also make a use as replacement of phosphates in sausage possible [19]. In recent studies, silver skin was used as a partial flour replacement [20,21] but was also used as an extract for its antioxidant properties [22]. As silver skin teas were described as rich in flavors [23] applications as food flavors are also possible. For use as extract, the process of extraction [24] and encapsulation [25] is also evaluated.

In the context of a consultation on the determination of the status of a novel food under Article 4 (2) of Regulation (EU) 2015/2283, the roasted silver skin of coffee beans (testa of *Coffea* sp.) was determined as a novel food. The rationale of the authority was that the largest share of roasted coffee beans marketed are not consumed directly. Only the hot water extract (coffee beverage) is used, while the insoluble parts (spent coffee grounds containing the silver skin) are usually discarded and not consumed (Figure 3). From the consumption of the hot water extract, a history of consumption of the isolated silver skin cannot be deduced. Furthermore, the consumption of silver skin as part of roasted coffee beans as such or in combination with nonfiltered coffee beverages cannot be taken as a history of consumption of isolated silver skin. The authority concluded that roasted silver skins of coffee beans have not been used for human consumption to a significant degree within the Union before 15 May 1997 [26].

As a novel food, the roasted silver skin needs a pre-market authorization before being placed on the market within the EU. The authorization procedure also includes a risk assessment by the European Food Safety Authority (EFSA). Considering the necessary authorisation procedure, this review article summarizes the available toxicological literature on coffee silver skin.

## 2. Materials and Methods

For this article, electronic searches of the literature were conducted using different databases including PubMed (National Library of Medicine, Bethesda, MD, USA), SciFinder${}^{n}$ (American Chemical Society, Columbus, OH, USA) and Google Scholar (Google LLC, Mountain View, CA, USA). A systematic search was performed for the years 2020–2022; for older publications, the research group already had a literature overview available during the previous review of Klingel et al. [7]. From these results, references were used to go back into the publication history for the topic, and authors with relevant publications were searched via Google (Google LLC, Mountain View, CA, USA) and Research Gate (Research-Gate GmbH, Berlin, Germany) to check their list of publications for further and more recent articles. The result was a narrative review aiming for completeness of available evidence.

A wide range of search terms was used, including coffee (restricted to toxicological hits), coffee silver skin, silver skin, coffee chaff (a term which is used for silver skin as animal feed), caffeine, chlorogenic acids, polyphenols, flavonoids and melanoidins. In addition, standard works of literature were used concerning knowledge of the product and toxicological relevance. Furthermore, databases such as the European Food Safety Authority (EFSA) database on food intake were searched for product groups and the Deutsche Gesellschaft für Ernährung (DGE) database was used for general knowledge of nutrition and mineral and vitamin intake, as well as the EFSA homepage for the terms coffee silver skin, caffeine, and phytosterol oxidation products.

## 3. Compositional and Toxicological Data on Coffee Silver Skin

The composition of coffee silver skin has been analyzed in several studies. A larger number of experiments using cell lines was performed. Additionally toxicological data from animal experiments are available.

### 3.1. Description of Coffee Silver Skin

This section gives an overview of the composition of macro and micro nutrients, and other plant secondary compounds, as well as known contaminants and residues.

#### 3.1.1. Macro Nutrients

The composition of coffee silver skin has been studied by various researchers. An overview of the major composition is given in Table 1.

Silver skin is a dry coffee by-product with a low moisture content of up to 7% [30,31,36]. This results in a raw material that is easy to store and transport being not prone to microbiological spoilage.

With up to 3% fats [3,27,28,29,30,31,32], silver skin can be considered a low fat food. Rancidity or oxidative fat degradation during storage have not been described, making silver skin also a stable material from a chemical standpoint. The main fatty acids are palmitic acid, linoleic acid, and behenic acid [27] with 64% saturated fatty acids in general, 30% polyunsaturated and 6% monounsaturated fatty acids. On the other hand, Nolasco et al. [35] report, with less detail than in the other sources, that from 3% fat only about half (1.2%) is saturated, depending on coffee species and country of coffee origin.

Proteins were found to be present with up to 22% [3,29,31] but according to a recent study by Machado et al. [37] a quarter of total nitrogen corresponds to the non-protein fraction. They calculated the amount of protein in the silver skin to be 12%, and by analysis found it to be 9%. Prandi et al. [33] support this by reporting a value of slightly above 7% protein. Regarding amino acid composition glutamic acid, aspartic acid and leucine are dominant. All essential amino acids were found in silver skin in free form except methionine [3]. The nutritional value of the proteins is 0.4–0.5, which is in range with most plant based proteins [33].

The fiber content is very high with up to 68.5% [28,29,31,34,35]. Main constituents of the fiber fraction are cellulose and hemicellulose with 51.50% and 40.45% respectively [29]. The fibers are classified in water soluble and insoluble parts with varying amounts according to method of preparation and analysis as given in Table 1. Low molecular weight soluble fiber was not detected in silver skin [38]. Looking at the contained monosaccarides of the fiber, xylose, arabinose, galactose, and mannose are the main ingredients [7]. Regarding carbohydrates, the sugars α-D-glucopyranose, β-D-glucopyranose, β-D-pyranose, β-D-fructofuranose, sucrose, and inositol were found to be the main constituents [39]. With up to 14.5%, the carbohydrate content is low [31].

In aqueous solution, the pH of silver skin extract is about 5, which is in the same range as coffee itself [40].

In summary, coffee silver skin can be assumed to be a dry low fat high fiber food. The protein content is probably high, but the assignment of the full nitrogen fraction as protein is uncertain.

#### 3.1.2. Vitamins and Minerals

Vitamins were found to be present with vitamin E at around 4.17 mg/100 g [36,41], which corresponds to approximately 1/12 of daily reference dose in 26 g. B-vitamins, mainly B2 with 0.2 μg/g and B3 with 3 μg/g [42], which are both less than 1% of daily reference dose in 26 g [43]. 26 g is the intake of silver skin from all sources, estimated by the intake of all potential food applications.

Minerals were widely analyzed in coffee silver skin. Potassium is present with about 5000 mg/100 g, magnesium with 200–2000 mg/100 g, calcium with 500–1000 mg/100 g and iron with 18–80 mg/100 g, zinc with 0.7–2.2 mg/100 g [29,35,36,44]. That are in 26 g of estimated daily intake of silver skin 32% of daily dose for potassium, 14–140% for magnesium, 13–26% for calcium, 50–200% of iron and around 5% for zinc.

#### 3.1.3. Plant Secondary Compounds

Coffee silver skin contains several plant secondary compounds, the most widely researched ones are listed in Table 2.

All samples were extracted with water or solvents to perform the analysis. The type of extraction was not standardized between different research groups, and it cannot be guaranteed that the total amount was found in the extract. Most contents are given for dry matter, deviations are flavonoids and chlorogenic acid which are given as amount in the extract.

Although there are different specific effects, all have in common that they have antioxidative properties.

The most expected ingredient for a coffee by-product, caffeine, is present with about 0.65–1% [3,31] in various extracts of silver skin. The wide distribution can be explained with the two different main coffee species with their numerous varieties. The amount of caffeine is comparable to the coffee bean. Caffeine itself is well known and well described in its physiological properties. The stimulation of central nervous system as well as the mild diuretic effect are the most common but many other effects are investigated such as cardio protective properties. Caffeine is also used in cosmetics against cellulite and hair loss [45,46]. On the other hand, there is an upper limit for safe daily intake, which is lower for pregnant women [50].

Another important component of coffee silver skin are polyphenols, including flavonoids. Polyphenols are secondary plant substances with many different functions in the plant, ranging from colorants to biopolymers. For different plants, the effects of polyphenols are known, among others, to reduce the risk of stroke, improve insulin resistance, as flavonoids may influence cardiometabolic health [51]. In silver skin, polyphenols are present with 1–6 mg gallic acid equivalent (GAE)/g and flavonoids with 0.18–2.35 mg rutin equivalent (RE)/g [47,48], which is below the range of known foods rich in polyphenols, such as blueberries [52]. Different definitions of polyphenols as well as different methods of analysis make comparison difficult. The total phenolic content, antioxitative capacity as well as the bioaccessibility of the polyphenols depends on the origin of the coffee [53]. Considering the amount of chlorogenic acid in silver skin extract, silver skin is nevertheless a food rich in polyphenols.

Chlorogenic acid is present with up to 6.8% in extracts of silver skin [49]. Antidiabetic as well as antiinflammatory effects are known and can be traced back to its antioxidative effects [45]. Other polyphenolic acids with less content are cinnamoylquinic acid and feruloylquinic acid including isomers [39]. Resveratrol was not detected [46]. Melanoidins are known to develop in roasted plant material, since silver skin is roasted together with the coffee bean, the content can increase by up to 23%. The structure of melanoidins is complex and not yet fully known. The investigated effects are, among others, anticancer and anticholesterol, but it is also known that the intake of trace elements is inhibited [47].

Trigonelline, a well known component of coffee, is also found in silver skin with up to 3.65%. It can form vitamin B3 during roasting [39]. Regarding phytosterols, β-sitosterol with 77 mg/g was found as dominant component [42]. Like coffee beans, silver skin also contains many odor active compounds. The composition of the odor mixture varies so that the aroma of silver skin extract differs from coffee and can be used to flavor different types of food, for example as a vegan smoke flavor [54]. Short-chain fatty acids with an assumed effect on the microbiome make silver skin a possible functional food [27].

#### 3.1.4. Contaminants

In addition to the desirable ingredients mentioned in the previous sections, possible contaminants of silver skin should also be considered.

Ubiquitous contaminants are heavy metals, mainly lead, mercury, and cadmium. In silver skin mercury and cadmium were found to be 0.05 mg/kg and 0.07 mg/kg respectively, lead content is up to 0.36 mg/kg with big differences between the samples from different sources ranging from <1 to 0.36 mg/kg [42,44]. A further study reports a lead content of 2.63 mg/kg [35]. There are no specific limits in the EU for coffee. In comparison for wheat, limits are 0.1 mg/kg for cadmium and 0.2 mg/kg for lead, 0.3 mg/kg lead is tolerable for leafy vegetables [55,56]. Heavy metals are taken up from ground, water and fertilizers by the plant. To achieve qualities with low content of heavy metal contamination, the growing conditions of the coffee tree have to be considered.

5-Hydroxymethylfurfural, a compound built by thermal decomposition of carbohydrates and suspected to be genotoxic and carcinogenic, was found with 0.57 mg/kg [36]. This is at least an order of magnitude less than a food rich in 5-hydroxymethylfurfural. Silver skin is not considered a relevant source of intake within the acceptable daily intake [57]. If needed, a mitigation would be possible by lower temperature or less roasting.

Furfuryl alcohol was not detected [3], polycyclic aromatic hydrocarbons were only detected in traces, furan, methylfuran, and pesticides were also below the detection limit [44]. Nolasco et al. [35] fount dibenzo-pyrene to be the main polycyclic aromatic hydrocarbon with 0.86 mg/kg, all other polycyclic aromatic hydrocarbon were found to be below 0.1 mg/kg. Benzo[a]pyrene was below the detection limit of 0.01 mg/kg.

These and further contaminants to be considered are shown in Table 3. Acrylamide and phytosterol oxidation products (POP) are roasting by-products; ochratoxin A and aflatoxins are mycotoxins, occurring in molded raw materials (i.e., improperly processed or stored coffee beans).

Acrylamide is found in a wide range from 11.42 μg/L in an extract up to 720 μg/kg [3,35,40,60] in silver skin dry matter. This is reasonable, as the degree of roasting can also vary in a wide range, and there can be various mixtures of the two species of coffee. However, the range measured from a set of diligently roasted samples of <20–161 μg/kg is below the benchmark level for coffee and cookies [61]. It is necessary to monitor and possibly mitigate this contaminant, but the average values are not in conflict with the food regulations.

POP were analyzed in different roasted varieties of silver skin and found to be 2.1–8.8 mg/kg. The dominant POP were identified as products of sitosterol, the phytosterol with the highest concentration in coffee silver skin. POP have exhibited adverse effects in a long term study and POP from sitosterol are cytotoxic. The intended applications of silver skin are not assumed to exceed or contribute significantly to the amount of 0.64 mg/kg bw/day, which was considered safe by EFSA [23,62]. Acrylamide as well as POP are roasting by-products. Their content could be lowered by carefully controlling the roasting conditions of the coffee bean.

Ochratoxin A, which can induce renal toxicity, nephropathy, and immunosuppression [65], was found in different studies in ranges from below 4 up to 34.4 μg/kg. The value is influenced by preparation and storage of silver skin, but also by roasting [63,64]. There is no limit for silver skin itself in Regulation (EC) No. 1881/2006 [59] but for coffee there is a limit of 5 μg/kg. The same limit applies for cereals.

Aflatoxins, known carcinogenic mycotoxins, were analyzed [34] but were well below the food limits, except for the limits for baby food and food for medical purposes according to Regulation (EC) No. 1881/2006 [59]. Aflatoxins and ochratoxin A are mycotoxins, occurring in molded raw materials. Their content can be kept low by properly storing and processing the raw coffee beans.

### 3.2. Use History and Intake

Coffee itself has been used for several centuries. Coffee silver skin as part of the coffee bean was removed more or less thoroughly from the bean, but always remained at least in parts. From traditional roasting in pans with almost complete retention to modern roasting with about 2% retention in coffee beans, there was always silver skin present in diet. The Italian coffee law [66] gives an upper limit for silver skin content, which shows that the presence was known. The aqueous extract of coffee beans has a long history of use. Direct consumption of coffee beans in Tiramisu or chocolated coffee beans is also known, but usually plays a minor role in the diet. Coffee silver skin has been used in animal feed for a long time, where it is considered safe at concentrations of up to 10% [18,38] and is listed in Regulation (EU) No. 68/2013 in the catalog of feed materials [67].

Recent applications are planning to add higher amounts of silver skin to the human diet. Several research groups partially replaced flour with silver skin, up to 30% in cakes [20], 6% in vegan biscuits [31] or 3% in biscuits [60]. In cookies an addition of 5% silver skin showed up as optimum [21]. In flat bread, up to 5% were used [68]. In addition to bakery products, up to 6% were added to yogurt [30] and 3% to hamburger patties [69]. Other conceivable options include the partial substitution of flour in other variants of bread and use in protein bars, fitness products or other sports products, as well as in energy bars [54]. A recent study by Thangavelu et al. [19], which is not further taken into account for intake estimation, gives a use concentration of 0.8% silver skin in sausages. Silver skin is used as texturing agent to replace phosphates.

The intake from possible sources is estimated in Table 4. The concentration of coffee silver skin refers to the concentrations given in the previous paragraph, the consumption of each foodstuff refers to the food consumption statistic of EFSA [70]. To estimate the upper intake limit, it is assumed that silver skin is added to all products of the respective product group. This approach also covers a certain amount of use of silver skin in products for which no use concentration is known yet.

Data on adolescents and children are used to generate an average intake of a larger population group. Due to differing caffeine intake limitations, children and pregnant women must be considered separately and are not evaluated as the target group for silver skin products. For an average adult person of 70 kg body weight (bw) the maximum worst-case total intake per day is assumed to be 26 g, when the diet is exclusively based on the mentioned silver skin products.

### 3.3. Adsorption, Distribution, Metabolism, Excretion and Nutritional Data

From the current state of research only limited information on nutritional data and metabolism are available for isolated silver skin. Comprehensive data are available for its main constituents such as caffeine or chlorogenic acid.

#### 3.3.1. Adsorption, Distribution, Metabolism, Excretion

With regard to the adsorption, distribution, metabolism, and excretion of coffee silver skin, limited information is available. The effects of silver skin on cells are discussed in Section 3.4.3 and animal studies in Section 3.4.5.

Tores de la Cruz et al. [71] investigated the effect of an extract of silver skin rich in fiber and melanoidins with little content of caffeine or chlorogenic acids. They showed that the fibers are excreted via feces and the animals’ cecum filled-up significantly faster. The number of fecal pellets increased due to the fiber effect.

Fernandez-Gomez et al. [72] showed that caffeine from silver skin extract is excreted via urine as well as metabolized to paraxanthine. Caffeine is excreted within 12 h, paraxanthine within 24 h. Chlorgenic acid is metabolized to hippuric acid. Hippuric acid is excreted via urine, with a peak within 2 h after administration.

The same authors showed that the different effects of caffeine or chlorogenic acid could be influenced (intensified or reduced) when used in silver skin extract compared to the same concentration of pure substance. The substances responsible for the change in efficacy are not yet known. On the other hand, caffeine and chlorogenic acid in combination have a stronger impact on biological effects than each substance alone.

#### 3.3.2. Nutritional Data

Coffee silver skin features a low fat and carbohydrate content combined with a relatively high content of proteins and a high fiber fraction. Possible applications of pure silver skin include high fiber and high protein foods, suitable for energy-reduced foods or high protein sports foods. Regarding proteins, it is unclear whether the total nitrogen fraction of up to 22% belongs to proteins or only in parts [33,37]. Anyhow, all essential amino acids except methionine are present which makes silver skin or its protein fraction suitable as protein rich ingredient in food.

All components and contaminants relevant for nutritional rating are given in Table 5.

The second application is supported by the fiber content. In the standard western diet, too little fiber is consumed but too many calories. Many of the possible applications listed in Section 3.2 aim at this point and intend to replace a certain amount of flour with silver skin to obtain food with reduced calories and higher fiber content.

Regarding vitamins, silver skin is not supposed to add a noteworthy amount to the diet. Its main vitamin is vitamin E, which would be consumed in 26 g silver skin in levels of around 1 mg/day or less, which equals to less than 7% of daily reference intake [43]. However, vitamin E deficiency is hardly relevant in the western diet.

With regard to bulk and trace elements, the result is different. Zinc has a calculated intake of 0.6 mg, which is up to 7.5% of the daily reference dose in 26 g silver skin. Similar to vitamin E, deficiency is rare in western countries. For potassium (1310 mg, approx. 32% of the reference value) and calcium (264 mg, 26% of the reference value), a notable amount of daily intake could be reached. Potassium is generally included in sufficient amounts in the diet and a higher intake of natural foods is not problematic. Calcium is generally consumed below the reference dose and the addition of silver skin could be of value specifically for people who do not consume dairy products [73].

For magnesium too little data is available to set a daily reference intake, so that the intake is estimated. Magnesium intake from silver skin products exceeds this value (300–350 mg) with up to 524 mg. Neither relevant deficiencies nor exceedances are known due to diet when kidneys are healthy. The calculated iron intake is 21 mg which reaches more than 100% of the daily reference dose. Usually with less than 10%, a lower fraction of vegetable iron is absorbed via the intestine compared to animal products. Despite the high iron content, it is therefore not likely to consume an excessive amount of iron except in cases of iron absorption disorders.

Aflatoxins were found to be below detection limit, which is below all limits except the ones for baby and medical purpose infant food. Aflatoxins are no contaminant of concern [34,59].

For POP, a daily intake of about 0.003 mg/kg bw per day is calculated for the assumed intake of 26 g silver skin, as the upper limit of the intake of silver skin added to foods. This intake does not contribute significantly to the limits of safe intake set by EFSA of 0.64 mg/kg body weight per day.

Ochratoxin A was detected with < 4μg/kg [63] but also up to 34.4 μg/kg [64]. Although the lower value is in accordance with grain limits according to Regulation (EU) No. 1881/2006 [59], the high values lead to the exceedance of the limits. This is also possible for compounded products when contaminated silver skin would be used as food ingredient. As a low ochratoxin A content is possible, this value should be monitored and should not exceed the limit of 5 μg/kg which corresponds to the limit for coffee beans or grains.

Acrylamide was found in silver skin with <20–161 μg/kg [3,44] as well as with 720 μg/kg [40]. One of the benchmark levels in Regulation (EU) No. 2017/2158 is the limit for wheat bread with 50 μg/kg. This necessitates the use of grades with low acrylamide content. For bakery products, there is a benchmark level of 300 μg/kg which would allow ad libitum doses of silver skin. Values around 720 μg/kg are clearly too high for any application in reasonable amounts. This level from the literature also needs to be questioned as it is in divergence from the other references, and common knowledge about acrylamide formation behaviour in coffee. As the silver skin is on the outside of the coffee bean, it is expected to be exposed to the highest levels of thermal energy leading to reduced levels of acrylamide [74].

Mercury was found with 0.05 mg/kg [42]. Although there is only a limitation for fish with organic mercury, plants contain inorganic mercury that is not well absorbed into the body. Therefore, the measured value is low and is not of concern and does not significantly contribute to TWI [58]. Cadmium was found with 0.07 mg/kg [42] which is in accordance with the limits for wheat and therefore also not of concern. The content of lead with up to 0.36 mg/kg [42,44] or even 2.63 mg/kg [35] is over the limit for grain which is 0.2 mg/kg. The value of 2.63 mg/kg is clearly too high for food use and therefore not included in further consideration. Given the use concentration of up to 7% in food the contribution from silver skin amounts to 0.025 mg/kg or an intake of 0.13 μg/kg bw per day from all silver skin products. Monitoring of this contaminant is recommended. However, this content appears to depend on the source of the silver skin, since silver skin without detected lead was also measured [44].

Given a caffeine content of 0.65–1% in silver skin, the calculated caffeine intake by all silver skin products is 170–261 mg per day. Although the lower value is in accordance with the limit for pregnant women of 200 mg/day, the latter value is only in accordance with the limits for adults, except pregnant women of 400 mg per day [50]. 7% of silver skin as the maximum amount used in cakes to receive healthier cakes with increased fiber content leads to a caffeine concentration of 700 mg/kg. Labeling indicating the presence of caffeine and warning of sensitive consumer groups are necessary.

There are no limits for other physiological active components melanoidins, flavonoids and polyphenols and chlorogenic acid. As these compounds are also present in coffee, which is consumed in larger amounts, no safety concerns are expected.

### 3.4. Toxicological Data

Several toxicological studies were performed with coffee silver skin and its extracts. Many of the studies were carried out with cell lines or invertebrates, but some studies with rodents are also available.

#### 3.4.1. Human Skin

Rodrigues et al. [75] performed in vitro skin and eye irritation tests. For skin irritation a reconstructed human epidermis, EpiSkin™, was used. For eye irritation SkinEthics™ HCE, a human corneal epithelial model was used. Both models are validated as replacements for animal tests, and the respective OECD guidelines (TG 492 and 439) are available and used for regulatory needs. Silver skin extracts were prepared with water, ethanol, and a 1:1 mixture of both. Neither for skin nor for eye irritation were any positive results found. Silver skin extracts are not irritating to the eyes and skin.

Additionally, a patch test was performed with 20 humans, the results are discussed in Section 3.4.7 human data.

#### 3.4.2. Acute Toxicology

Beltran et al. [34] performed an acute oral toxicity test with aqueous coffee silver skin extract according to OECD 425. The animals were administered a single dose of 2000 mg/kg bw by esopharingeal cannulation. The animals were observed 14 days after administration. No changes in behavior or food and water consumption were found. There were neither clinical signs nor deaths during observation period. LD50 is higher than 2000 mg/kg bw.

#### 3.4.3. Genotoxicity and Other Toxicological Relevant Cell Lines

Several studies were performed using different cell lines. An overview is given in Table 6.

Iriondo-DeHond et al. [65] investigated the cytotoxic effect of silver skin extract on human hepatocellular carcinoma cells (HepG2). An aqueous extract with 50 g/L silver skin was prepared by boiling for 10 min and filtration afterwards. Cell viability was determined by the MTT [3-(4,5-dimethylthiazol-2-yl)-2,5-diphenyltetrazolium bromide] assay with up to 20,000 μg/mL coffee silver skin extract for 24 h. Afterwards, cells were labeled, and optical density was examined. There was no effect up to 10,000 μg/mL silver skin extract, which was defined by more than 80% viability. There was a significant decrease at 20,000 μg/mL, IC50 was calculated to be 13,158 μg/mL. Based on these findings, the authors performed a comet assay to test DNA damage by coffee silver skin extract. HepG2 were exposed to 1–1000 μg/mL coffee silver skin extract for 24 h. The comet score was considered a parameter for DNA damage from strand breaks (no enzymes added) or oxidative DNA damage in purines or pyrimidines (test-specific enzymes added). While the positive control showed positive results, the test was negative for all concentrations of silver skin extract. Furthermore, coffee silver skin extract was shown to significantly reduce DNA damage caused by benzo[a]pyrene.

Rodrigues et al. [76] also investigated the cytotoxic effects of coffee silver skin extracts. They used 1 g of silver skin in 20 mL of water, ethanol or a 1:1 mixture of both, heated for 30 min on 40 °C and filtered. Human immortalized non-tumorigenic keratinocyte cell line (HaCaT) and human foreskin fibroblasts (HFF-1) were used as cell lines tested. Cell lines are suitable models for dermal contact applications. As indicators of viability uptake of a mitochondrial dye and leakage of cytosolic enzyme, lactate dehydrogenase (LDH) in the cell medium (LDH assay) was used. There was no cytotoxicity against HaCaT cells at all concentrations up to 1000 μg/mL silver skin extract. There was no effect on fibroblast viability for all extracts. LDH assay showed no effect in all concentrations tested for both cells types. No cytotoxic effect was observed.

Juan-García et al. [77] investigated the protective effect of silver skin extracts against the cytotoxicity of mycotoxins. 1 g and 10 g of silver skin were extracted with boiling water, alcohol (MeOH), and hydro-alcoholic mixtures: MeOH:H${}_{2}$O (*v*/*v*, 70:30), MeOH:H${}_{2}$O (*v*/*v*, 50:50), and EtOH:H${}_{2}$O (*v*/*v*, 70:30), the mixtures were concentrated to a volume of 50 mL. The MTT assay was used to determine cell viability, the test strain was the human neuroblastoma cell line (SH-SY5Y). Dilutions from 1:0 to 1:16 were used to investigate toxic effects of the extracts. Silver skin extracts showed increased viability of cells compared to control. The cytoprotective effect of silver skin extract against the mycotoxins α-zearalenol and beauvericin was investigated. SH-SY5Y cells were pretreated with aqueous extract after the administration of mycotoxins. Cell viability decreased compared to α-zearalenol alone, for beauvericin at some concentrations a cytoprotective effect was found. Simultaneous treatment of silver skin extract with mycotoxins showed protective effects for both mycotoxins.

Costa et al. [36] investigated the effect of aqueous and ethanolic silver skin extracts on erythrocyte oxidative induced hemolysis. 1 g of silver skin was extracted with 50 mL water or ethanol. Erythrocyte suspensions with 2.6% hematocrit were prepared from human blood. Hemolysis was induced by 2,2´-azobis(2-amidinopropane) dihydrochloride or hydrogenperoxide. Additionally, the oxidative status of hemoglobin and erythrocyte morphology was evaluated. The advantage of erythrocytes is that they have poor mechanisms of repair and biosynthesis, oxidative stress cannot be compensated besides a certain amount of contained antioxidants. Therefore, an excess of oxidative stress leads to membrane defects followed by hemolysis. The extracts showed a significant protective effect which was dose dependent. 25 g/L had a significant effect, at 0.2 and 0.78 g/L there was no effect for the different inducers of oxidative stress. Effects at higher concentrations, 50 and 100 g/L, were assessed optically as the solutions were too dark for spectroscopic evaluation. At those high concentrations, the effect turns into the opposite, the antioxidant capacity disappears, and a pro-oxidant effect shows up. Up to 25 g/L, silver skin extracts have no oxidative effect on hemoglobin.

Tores de la Cruz et al. [71] showed that silver skin extract reduced reactive oxygen species (ROS) in rat small intestinal epithelial cells (IEC-6). The extract preparation is the same Iriondo-DeHond et al. used [65] and additionally a melanoidin rich fraction was prepared from the residues of extract generation. Cytotoxic level was evaluated and noncytotoxic levels (0.004, 0.04, and 0.4 mg/mL) were chosen to investigate their effect on cell redox status. Extracts significantly reduce intracellular ROS production, silver skin extract in concentrations of 0.04 and 0.4 mg/mL is comparable to 10 μg/mL vitamin C, which is an antioxidant control. When intracellular ROS were induced by tert-butylhydroperoxide, silver skin extract showed a reduction in ROS for all concentrations. The extract containing the melanoidin rich fraction showed a reduction in ROS at the highest dose level.

Fernandez-Gomez et al. [78] showed that silver skin extract is able to modulate the insulin secretion of pancreatic rat insulinoma cells. The aqueous silver skin extract was prepared with 2.5 g silver skin in 50 mL water, 1–10 μg/mL were used as dose. Chlorogenic acids and caffeine were found to be present in the extract with 11.18 mg/g and 30.26 mg/g respectively. Caffeine and chlorogenic acid were also tested separately with 1–10 μM to assess whether they are the active components of the silver skin extract. None of the substances affected the intracellular generation of reactive oxygen species or cell viability. Regarding glutathione peroxidase (GPx) and glutathione reductase (GR) activities, any of the substances preserved glutathione storage and GR activity, and silver skin extract and chlorogenic acid increased GPx activity. These enzymes are part of the defence against peroxides and superoxides, phenolic compounds have already shown to enhance enzyme activity. Pretreatment with silver skin extract and chlorogenic acid increased insulin secretion in cells cultured in media containing 4 and 10 mM of glucose. Caffeine was ineffective. Silver skin extract was effective in lower doses (1 μg/mL), the resulting concentration of below 0.0001 μM chlorogenic acid was ineffective. By measuring the insulin content, the authors showed that the increased insulin secretion was not related to increased biosynthesis of insulin but to stimulation of secretion. Streptozotocin, a DNA methylating agent generating reactive oxygen species and thus oxidative stress, showed to decrease cell viability with increasing concentrations. 5 mM for 6 h was chosen to investigate the protective effect of silver skin extract, 90 min for insulin secretion assays. The cells were pretreated with 1 μg/mL silver skin extract or 10 μM caffeine or chlorogenic acid. Although streptozotocin induces morphological chances that are related to cell death and increased generation of reactive oxygen species, silver skin extract and chlorogenic acid, but not caffeine, were shown to be effective in protecting cells. In line with these effects, streptozotocin reduces insulin secretion and GPx and glutathione activities and chlorogenic acid and silver skin had a protective effect, caffeine had not. The protective effect of silver skin extract is believed to be caused not only by the content of chlorogenic acid, but also by other components of the extract.

Barreto Peixoto et al. [79] investigated the effect of silver skin extract on sugar intake. The extract was prepared with 0.5 g silver skin in 25 mL water or a multiple thereof in the laboratory or pilot scale with some difference in extraction methods. The cell line was human intestinal epithelial cells (Caco-2). The intake of ${}^{3}$H-Deoxy-D-Glucose (^3^H-DG) and ${}^{14}$C-Fructose (^14^C-FRU) by Caco-2 Cells was measured as well as the amount of SGLT1 (sodium/glucose cotransporter 1), GLUT5 (fructose transporter) and GLUT2 (glucose transporter 2) mRNA normalized to the amount of human β-actin mRNA, a housekeeping gene. In addition, cell viability and mass were measured. All extracts were able to reduce ^3^H-DG uptake into Caco-2 cells significantly at 1 mg/mL, one of the extracts reduced it by 50%. ^14^C-FRU was reduced by 2 of the 3 extracts. Pure caffeine and chlorogenic acid were studied separately and combined as a reference. The concentration was chosen according to the concentrations in the silver skin extracts. Caffeine and chlorogenic acids did not inhibit sugar intake, when combined they reduced ^3^H-DG uptake. The inhibition was not the same as for the extracts, which shows that further components in the extract are involved in inhibition. Caffeine reduces ^14^C-FRU uptake, and chlorogenic acid shows a synergistic effect. Together, both components reduce uptake in the same way as silver skin extracts; 2 of the extracts reduce less than the mixture of caffeine and chlorogenic acid in the same concentration present in the extracts. Probably, further components of the extracts reduce the efficacy of caffeine and chlorogenic acid. Regarding glucose and fructose transporters, 24 h of treatment with the extracts reduced the expression levels of GLUT2 to around 30% of the control, SGLT1 to approximately 55% of the control, and had no effect on the expression of the GLUT5 gene. The reduced expression can be related to reduced uptake from the previous experiment. The extracts were not cytotoxic at 1 mg/mL, but a reduction of 16–24% in culture mass was observed. The reduction in sugar intake was not due to cytotoxicity, but an antiproliferative effect could not be ruled out.

Rebollo-Hernanz et al. [80] investigated adipogenesis-related inflammation, mitochondrial dysfunction and insulin resistance using adipocytes. As cell lines, mouse 3T3-L1 preadipocytes and RAW264.7 Mϕ, murine-leukemic monocyte-macrophage cell line, were used. Phenolic rich extracts of silver skin were produced by boiling 25 g silver skin in 500 mL water for 10 min. Chlorogenic acid and caffeic acid represented 97% of polyphenolic compounds in the extract with 2.8 mg/g and 0.5 mg/g respectively. The accumulation of lipids in the adipocytes was decreased by the extract, and the release of glycerol in the mature adipocytes increased. The same effect was shown for the pure polyphenols (caffeic acid, chlorogenic acid, protocatechuic acid, gallic acid kaempferol) that were tested as a reference. Lipase activity as well as citrate synthase activity and ATP production were increased by silver skin extract.

Coffee silver skin extract reduced lipopolysaccharide induced inflammation and Mϕ activation, but pure polyphenols showed better results. According to Tores de la Cruz et al. [71] a reduction in ROS was found. Kaempferol and chlorogenic acid were identified as the compounds in the extract that led to this effect, kaempferol being the most effective substance. Reductions of tumor necrosis factor α release in adipocytes which were treated with macrophages culture medium and the extracts and phenolic compounds. Silver skin extract and pure phenolic compounds reduced adipokine secretion caused by culture medium. Peroxisome proliferator-activated receptor gamma coactivator 1-alpha (PGC-1α) expression was restored by silver skin extract as well as the pure phenols.

According to the results of Fernandez-Gomez et al. [78] it was shown that silver skin extract modulated the phosphorylation of a large part of the proteins involved in the insulin receptor (INSR) signal transduction pathway, increasing insulin dependent glucose uptake. The effect of pure phenolic compounds (chlorogenic acid, protocatechuic acid) was greater.

#### 3.4.4. Other Cell Lines

Rodrigues et al. investigated the antimicrobial activity of coffee silver skin extracts [76]. Silver skin extracts were prepared with water, ethanol, and a 1:1 mixture of both. Tested microorganisms were *Staphylococcus aureus*, *Staphylococcus epidermidis*, *Escherichia coli*, *Klebsiella peumoniae*, *Pseudomonas aeruginosa* and *Candida albicans*. There was no effect against *C. albicans* and *P. aeruginosa* but for the others inhibitory effects were found at 250 μg/mL or less silver skin extract incorporated into culture medium.

Prandi et al. [33] showed for simulated stomach conditions (feed suspension, pH 3.5, 37 °C) using *E. coli* and *Streptococcus suis*, that the extracts caused an increase in antimicrobial activity but showed effects against *Streptococcus suis* at the higher dose of 1.25 kg/MT.

Ozmen-Togay et al. tested the effect of coffee silver skin on kefir cultures of lactic acid bacteria as well as the effect on their survival in a simulated gastrointestinal tract [81]. Coffee silver skin of *C. arabica* and *C. canephora* was added at concentrations up to 1% at the beginning of fermentation. The *C. canephora* silver skin increased the viability of lactic acid bacteria in kefir after storage, and both types of silver skin increased viability after the in vitro gastrointestinal tract.

#### 3.4.5. Chronic Exposure

Several subchronic studies were performed. One study aimed to investigate the subchronic toxicity of silver skin, the other studies aimed to show special effects but are nevertheless useful to evaluate the (sub)chronic toxicity of silver skin. An overview is given in Table 7.

Iriondo-DeHond et al. [82] performed a subchronic toxicity study with rats receiving 1 g/kg bw/day of silver skin extract for 28 days. No adverse effects were observed. The extract used was an aqueous extract of 50 g/L silver skin (*Coffea arabica*), 10 min at 100 °C, filtered, and finally the filtrate was freeze-dried. 14% of the initial sample were received as an extract. As a limitation of the experiment, the non-water-soluble parts of the silver skin are lost for the toxicity test. Those parts are expected to be mainly insoluble fiber. The study was carried out with a dose group and a control group with a total of 15 male and 15 female rats, 3 weeks old at the start of the study. The protocol used was the OECD Test Guidelines 407 (Repeated Dose 28-day Oral Toxicity Study in Rodents). In the control group, there were seven animals of each sex, in the dose group eight. Body weight, food and water intake, behavior and signs of toxicity were recorded daily throughout the study. Detailed clinical signs were recorded once a week for both groups. After the end of the study, tissue samples were investigated from the brain, lungs, liver, heart, spleen, thymus, kidneys, adrenal glands, and sex organs. Blood samples were taken from anesthetized animals and numerous parameters were analyzed, including hormone levels, oxidative stress biomarkers (non-enzymatic antioxidant capacity and enzymatic oxidative stress biomarkers), as well as inflammatory biomarkers (C-reactive protein (CRP)). Furthermore, the effect of dissolved fiber in the extract on dietary fiber was investigated by the number and weight of stool per 24 h, as well as pH, antioxidant capacity, and short-chain fatty acid content in the stool.

There was no significant change in food intake, but males showed slightly higher food intake and therefore increased fiber intake. Water consumption did not show any difference for the first 3 weeks; in the fourth week, female rats in the treatment group consumed more water than in the control group. No changes in organ weight were observed, there were no gross findings. The body weight of male rats was higher, which was in agreement with the higher food intake. Hemorrhage was found in the lungs of treated and control animals, Iriondo-DeHond et al. are leading this back to the method of sacrifice. In histopathology, no effects of silver skin parameters in the blood samples, also no differences between treated and control group were observed. There was also no effect on insulin, serotonin, and melatonin levels. Although antioxidative effects were found in vitro or for simpler animal models (*C. elegans*, see Section 3.4.6) there was no effect observed on total antioxidant capacity and enzymatic oxidative stress biomarkers or the inflammation biomarker CRP in vivo. Regarding feces, no change in number or weight was found in female rats, and for males with a higher food intake, an increase in the number of feces was found. There was no effect on feces weight, pH, or antioxidant capacity. Short-chain fatty acids were found in a higher amount in the feces of treated males.

Tores de la Cruz et al. [71] investigated the effect of *C. arabica* silver skin extract melanoidins on 14 male rats. Extract preparation was the same as that of Iriondo-DeHond et al. The melanoidin rich fraction was generated from the extraction residue, separated by ultrafiltration, and extracted once again with water, and finally the fractions were separated. The fraction is rich in melanoidins and contains slightly less protein and almost no caffeine or chlorogenic acid. It contains more fiber than the standard extract. Eight rats were in the control group, six in the melanoidin group, receiving 1 g/kg bw/day by drinking water Monday to Friday for 28 days. In the last week, gastrointestinal and colon mobility parameters, as well as dehydration, perianal area appearance, and fecal pellets were evaluated. There were no significant differences in body weight or food or water intake between the test and control groups. There were slight differences that are not statistically significant, the animals in the test group showed lower body weight due to lower food intake. This could be explained by the fiber content of the melanoidin extract. No signs of dehydration were found, perianal areas and feces were normal. The small intestine has a lower weight in treated animals, and the kidney and liver have a slightly lower weight. The effects do not sum up to a relevant toxicity. No differences in the lengths of the small intestine and colon were found. The caecum filled significantly faster for the treated animals, but when full, there were no differences between the groups. No differences were found for the stomach, but caecum was filled earlier, not due to faster gastric emptying or shorter small intestine but probably due to increased propulsive activity of small intestines. The number of fecal pellets was slightly higher for the treated group when measured after 4–8 h in the colorectum. There was no difference in fecal density, but the pellets were larger for the treated group. This effect might be related to a higher fiber intake. The total time until expulsion of a bead added to the colorectum showed no difference between the treated and control groups.

Kim et al. [83] investigated the effect of silver skin extract on muscle growth. They administered the extract to mice for 29 days orally. The study was carried out with 12 mice, each six in the test group and the control group, 6-weeks old at the beginning of the test. The mice received 150 μg of an ethanolic extract of coffee silver skin per day, body weight was measured weekly. On day 29 after the grip strength was measured and after 12 h of fasting, the animals were sacrificed. Blood was collected and muscle and bone weight was measured. For treated mice, the forelimb muscle mass was higher, the hindlimb muscle mass did not show difference. The grip strength of the treated animals was higher on days 22 and 29, but not earlier. There was no effect on bone and body weight. Regarding blood samples, the level of free fatty acids was lower in treated mice, but there was no effect on glucose, total cholesterol, and triglyceride. The effect of the extract on the mRNA genes related to energy metabolism was also investigated. For Ppargc1a and Ucp1, an up-regulation was found for the animals in the test group, and for Fndc5 and Mstn no effect was found. Ppargc1a is the gene for the protein peroxisome proliferator-activated receptor gamma coactivator 1-alpha, the main regulator of mitochondrial biogenesis and liver gluconeogenesis; Ucp1 relates to thermogenin, a mitochondrial carrier protein from brown adipose tissue which generates heat by non-shivering thermogenesis. Fndc5 encodes a membrane protein, Mstn encodes myostatin which causes muscle growth. The observed effect on muscle growth and increased strength in mice treated with coffee silver skin extract might be related to the polyphenol content of the extract [86].

El-Anany et al. [84] investigated the hypolipidemic effect of silver skin in rats fed a high fat diet. Unlike the studies described above, rats received untreated silver skin instead of extract. In total, 40 male rats were used in five groups, which means eight male rats per group. The groups received normal diet, high fat diet, and high fat diet with 10%, 15% or 20% coffee silver skin for 8 weeks. Serum total lipid level and total cholesterol concentrations, high and low density lipoproteins, and triglycerides were measured. Weights of hearts, livers, and kidneys were assessed, as well as epidermal and retroperitoneal fat. Liver and kidney were histopathologically examined. Activity of alanine-, aspartate aminotransferase and phosphatase were determined, as well as the urea and uric acid concentration from serum.

The rats gained more body weight with the high fat diet than the rats with the standard diet. There was no effect on food consumption between the 5 different diets. The addition of silver skin reduced body weight gain; the more silver skin was added, the greater the reduction observed. However, the weight gain still exceeded that of the control group. The effect of silver skin might be related to the high fiber content. Regarding organ weights and body fat, there was no difference in heart weights between all groups. The control group showed the lowest weights of liver and kidney, as well as the smallest amounts of body fat. High fat diet for 8 weeks increased the weight of liver and kidney, as well as body fat. The addition of silver skin reduces organ weights, as well as body fat, 15% and 20% had a significant effect. Liver and kidney are almost in the range of control group and body fat is only slightly higher. The high fat diet resulted in higher activity of kidney and liver enzymes, as well as increased amounts of urea and uric acid. The addition of silver skin reduced these activities and concentration; the more silver skin added, the greater the reduction. The same was detected for organ weight, with the addition of 20% silver skin there was not much deviation from the control group. Regarding blood serum, a high fat diet results in an increased concentration of all lipids except high-density lipoproteins, which decreased. Like for the previous parameters, the addition of silver skin decreases the lipids and increases the high-density lipoprotein, which results in an improvement of overall parameters. The more silver skin added, the more improvement was observed. In histopathology rats with high fat diet showed steatosis, and fat droplets were accumulated within hepatic parenchymal cells. Adding 20% of silver skin to the diet resulted in almost normal hepatic lobules. Smaller amounts of silver skin had partial effects. Regarding the kidney, a high fat diet resulted in swollen tissue with minimal changes (lipoid nephrosis). The addition of 10% silver skin resulted in a slight improvement, 15% and 20% silver skin lead to an organ comparable to the control group with a standard diet.

Fernandez-Gomez et al. [72] evaluated the bioavailability of silver skin extract and the bioactivity of coffee silver skin extract in the pancreas of streptozotocin–nicotinamide diabetic rats. The extract was prepared by boiling 2.5 g silver skin in 50 mL water. The doses applied provided 0.150 and 0.434 mg/d of chlorogenic acid and caffeine, which corresponds to moderate coffee consumption. The upper limit possible for silver skin extract intake was defined by its caffeine content (maximum 300 mg/day). Twelve 6-week old rats were divided into four groups, one control, and one group each for treatment with caffeine, chlorogenic acid and silver skin extract. In the morning, the silver skin extract group received a single dose containing 2.2 mg caffeine/kg bw, the chlorogenic acid group 1.5 mg chlorogenic acid/kg bw and the caffeine group 5 mg/kg bw. Urine samples were collected each hour for 6 h and frozen until analysis. The experiments were repeated after 3 days of clearance with the same animals. Urinary creatinine was measured as well as chlorogenic acid, caffeine and related compounds, hippuric acid, and paraxanthine. Chlorogenic acid was not found as such in urine because it is rapidly metabolized to hippuric acid. The highest level of hippuric acid excretion occurs within 2 h after administration. The amount is higher after administration of chlorogenic acid than after administration of silver skin extract or caffeine. Nonmetabolized caffeine was excreted after consumption of silver skin extract and pure caffeine, and excretion was almost completed in 12 h. The excretion of the caffeine metabolite paraxanthine was higher after the intake of pure caffeine, and the excretion took 24 h for silver skin extract and caffeine. For the following investigations of the effect on oxidative stress biomarkers, diabetic rats with a blood glucose level above 200 mg/dL were used. The animals were treated with silver skin extract or caffeine or chlorogenic acid for 35 days. Chlorogenic acid and caffeine showed protection against streptozotocin induced damage and protein carbonyl content decreased by 24% and 22% in the pancreas. Silver skin extract showed no protective properties against streptozotocin induced damage. Silver skin extract and chlorogenic acid reduced glutathione depletion in the pancreas of diabetic rats.

Del Castillo et al. [85] state that silver skin extract reduced the total cholesterol and triglycerides plasma levels in rats when treated for 45 days. Also inhibition of pancreas lipase, a key enzyme for fat digestion was observed while caffeine was ineffective. No further details are available for this study, but they align with El-Anany et al. [84].

In addition to toxicological studies, coffee silver skin is used in animal feed up to concentrations of 10% [18]. Long term negative effects are not known.

To sum up the subchronic studies, it can be said that coffee silver skin was tested in aqueous and ethanolic extracts, as well as untreated as such. No toxic effects were observed, and silver skin reduced the negative effects induced by the high fat diet. Most in vitro effects were not found in vivo. There are no safety concerns regarding coffee silver skin.

#### 3.4.6. Other Animal Data

Thiligene et al. investigated the effect of aqueous coffee silver skin extract on the root knot nematode *Meloidogyne incognita* [87]. The extract was shown to reduce the nematode population, but was less effective than a nematicide.

Osimani et al. found larvae of *Hermetia illucens* fed with added coffee silver skin to have a higher microbial count than larvae fed without [88]. However, this result is interesting and needs to be replicated, but is not relevant to human risk assessment.

Iriondo-DeHond et al. [89] as well as Martinez-Saez et al. [22] investigated physiological effects of silver skin using *Caenorhabditis elegans* as model organism. Iriondo-DeHond et al. investigated the antioxidant properties of aqueous silver skin extract. The survival rates of *C. elegans* after oxidative stress induced by UVC radiation were measured. Silver skin extract was added to culture media in several concentrations, the nematods were treated with UVC 45 s per day which resulted in a viability decrease. 1 mg/mL silver skin extract showed an increase in nematode longevity comparable to vitamin C.

Martinez-Saez et al. [22] investigated the effect of coffee silver skin on body fat in *C. elegans*. Extracts of coffee silver skin as well as pure caffeine and chlorogenic acid were incorporated in culture media of the nematodes. They were fed till young adult stage and total lipid contend was determined. Pure chlorogenic acid and caffeine showed a statistically significant dose-response effect reducing body fat accumulation. In the model studied, chlorogenic acid had a stronger effect than caffeine. In addition, the same effect was found for different silver skin extracts in a range comparable to the chlorogenic acid and caffeine content in the extracts. The extracts contain physiologically active amounts of the compounds, effects of other components could be ruled out.

#### 3.4.7. Human Data

Rodrigues et al. conducted a patch test with 20 volunteers to overcome the uncertainties in the network of pilo-sebaceous units and the hydrolipid film of their in vitro skin irritation tests in vitro tests [75]. The patch tests were performed with an extract of silver skin in water and ethanol in 1:1 ratio. The patch test showed slight erythema on two volunteers 2 h after patch removal, but at time of statistical analysis there was no difference to negative control. Furthermore, there was no effect on transepidermal water loss. The extract is safe for topical use.

In a further study, Rodrigues et al. [14] formulated a cream containing coffee silver skin extract and tested the effect on skin improvement. A cream containing silver skin extract was compared to a cream containing hyaluronic acid in 20 volunteers. Regarding skin moisture, elasticity, and wrinkle depth, there were no differences between formulations within 28 days, there was an improvement in both cases without statistically significant differences. The cream was well tolerated.

#### 3.4.8. Allergenicity

Coffee is a food with a long history of use. Thereby, silver skin as part of the coffee bean was always consumed, not in high concentrations but above the threshold for allergic reactions. Coffee is not known to be a food with an increased risk of allergies. A cosmetic raw material containing silver skin extract is also commercially available and not known to cause contact allergies. It is not likely that coffee silver skin will cause allergies in more than individual cases.

## 4. Discussion

Since coffee silver skin was officially determined as a novel food during a consultation request [26], the safety when used as a food must be assessed considering its composition and toxicological information.

Fungal toxins formed by mold can be found in silver skin, but only in low amounts which do not affect safety. This is not unexpected, as silver skin is a dry product with typically unfavorable growing conditions for molds and other microorganisms. As certain batches could be nevertheless contaminated, limits as well as monitoring is recommended for food use.

Iron was detected in large amounts, so that the estimated worst-case intake would exceed the daily reference intake. The amount was measured by plasma atomic emission spectrometry and atomic absorption spectrometry after ashing or digestion with nitric acid, for one sample [29,36] or for ten samples [44] where lower values for *C. arabica* were reported. Both methods can be to resumed as valid. While there can be deviations in the exact value, the two values examined by different authors and different methods are quite similar. So the range of iron content is plausible despite the extremely high content. The iron content is in the range of known iron rich foods (liver, sesame), but further check of iron content seems reasonable. Although with a lower iron content of below 20 mg/100 g and an assumed intake of silver skin of 26 g per day the recommended daily iron intake is not exceeded, it is with a higher iron content of up to 80 mg/100 g. However, it is iron from a vegetable source, which means it is available as Fe${}^{3+}$. While the recommended daily intake assumes an intake of 10–30%, vegetable iron per se is less absorbed, and many secondary plant compounds further inhibit iron intake. An excessive intake of iron from natural sources is not known; therefore, no risk to silver skin is expected. Nevertheless, silver skin is not suitable for persons with iron intake disorder.

Lead is present with <1–0.36 mg/kg [42,44]. The analysis can be rated as valid. A representative coffee silver skin sample provided by a large coffee trading company was used in one case, different coffee samples were used in the second study. The analysis was done triple or tenfold and performed by inductively coupled plasma - mass spectrometry (ICP-MS). The calculated intake of lead from silver skin from all sources is at maximum 9.4 μg per day, which calculates as 0.15 μg/kg bw per day. This can contribute to up to 1/3 of the average daily lead intake in the EU [90]. As negative consequences cannot be excluded for this intake, a monitoring and possible reduction of lead in silver skin is considerate. This could be done by chelating agents [42]. On the other hand, samples with less than 1 mg/kg lead, which was the detection limit, were found [44], so choosing low lead qualities is certainly preferable.

With up to 1% the caffeine content in silver skin is relatively high. Although the intake from all sources is calculated as safe for all adults except pregnant women, cakes with up to 700 mg/kg caffeine could have negative effects if over-ingested. If silver skin is used as such, a labeling indicating the presence of caffeine appears to be needed.

To overcome the caffeine content for applications that only focus on the fiber content, decaffeinated silver skin is an option, as well as performing an aqueous extraction before using the silver skin residue. Tores la Cruz et al. [71] showed that the aqueous extract contains the largest part of caffeine. The extract could be used for other applications that target dissolved secondary plant compounds. The caffeine content could be kept lower in the final product due to the lower amounts of use of the extracts.

Negative effects on the food were not found, neither the properties of cakes and cookies have deteriorated significantly nor a negative influence on yogurt was detected.

Promising antioxidant effects were found in cell cultures and nematodes as simple model organisms. But these results should not be overrated, as the metabolism of natural substances is very complex, and the effects are not shown in studies with higher animals. Anyway, no cytotoxic effects are found in concentrations relevant for the planned applications.

Regarding toxicology, many of the required studies were already performed. Eye and skin irritation as well as acute toxicity were performed as studies following the OECD guidelines. These studies can be assumed as valid and show no negative effects of silver skin.

Several subchronic studies are available, but no chronic study. While none of the subchronic studies fulfills all criteria of a guideline-study (OECD or similar), they all consistently show that silver skin has no negative effect on the test animals.

The studies used various silver skin extracts as well as silver skin itself, also in high doses without signs of toxicity. In general, an absence of toxic effects were found. The 1/5 silver skin content in the diet was shown to almost completely balance the negative effects of high fat nutrition. This also coincides with the fact that silver skin was found to lower cholesterol and triglycerides in the blood. Although the study that stated the effect on blood lipids was not available in detail, the results fit the other evaluated studies. Antioxidative effects could not be shown in vivo but on the other hand, no influence of caffeine on the sleep rhythm of rats was found.

The effects observed for silver skin and its extracts can be attributed in part to caffeine, chlorogenic acids, and other polyphenols, but not completely, indicating that additional secondary plant compounds and interactions of the complex mixture play a role. While further research in this topic is needed to investigate further applications, it is not necessary for safety assessment.

As none of the subchronic studies shows any evidence of toxicity, toxic effects in chronic studies are also unexpected. Given the long use history of small amounts of silver skin in human nutrition and the use as animal feed without toxicological concerns as additional information, no chronic studies appear to be justifiable.

An in vitro comet assay is available with negative results, no further tests were performed. This results in a formal data gap on mutagenic effects. Standard data requirement would include an Ames test to check for point mutations and a micronucleus test to investigate chromosome aberration. Regardless of the formal requirements, a full testing approach seems partly exaggerated. In all cell lines, low cytotoxicity or even cytoprotective effects were found when investigated.

Taking into account the composition of silver skin, mutagenic effects are most likely to be triggered by secondary plant compounds. For the single components, test results on mutagenic potential are available. Caffeine showed mutagenic properties in cells but not in mammalians [91] and EFSA evaluated safe intake values [50], kaempferol also showed an absence of adverse effects [92,93]. For chlorogenic and caffeic acid [94] antimutagenic effects were found, and for protocatechuic acid [95] negative results of the mutagenic potential are available. For coffee itself, in summary, negative results were found [96] in animals, but not in the in vitro tests. In summary, mutagenic effects in cell lines are probable for silver skin. However, these results should not be overestimated, as the composition of silver skin secondary plant compounds is very similar to the composition of coffee secondary plant compounds, and for coffee, no mutagenic potential could be shown. Adding the centuries-long use of coffee (including small parts of silver skin) with no known negative effect on reproduction or cancer formation, a mutagenic effect of silver skin seems very unlikely.

This indicates that there is no mutagenic potential and a full testing is not necessary. An Ames test could be performed to close the formal data gap. However, the results should not be overrated, and further testing of cell lines is not assumed to generate more relevant information.

Regarding reproductive toxicity and carcinogenicity studies are also missing. As there are no indications for toxicity on reproduction organs in the subchronic studies and no formation of tumors was observed in animal feed uses.

Again, the long use history as animal feed without known negative effects can be taken into account to rule out carcinogenic effects. Additionally, the small amount present in roasted coffee beans has a long history of use in human beverages. As carcinogenic effects usually have no threshold value, possible carcinogenic effects of silver skin should be noticeable among coffee consumers. Although there is no proven connection of coffee with cancer, a protective effect against liver cancer is known [97]. Similarly to chronic studies, conducting studies on carcinogenic or reproductive effect is neither necessary nor justifiable.

In summary, no toxic effects of silver skin were found or are expected. While the studies on mutagenicity formally represent a gap, no effect is expected. There is no indication to conduct studies on reproduction, carcinogenesis, or other chronic effects.

The reasoning for the classification as a novel food is, on the one hand, understandable, since there is no information that silver skin was ever consumed as such in greater amounts, except some evidence that it was used as a substitute for coffee during World War I [98]. On the other hand, this is true for spent coffee ground as well, which was rated as “not novel”. Considering that the silver skin remained completely on the coffee bean until industrial roasting processes were introduced, treating the silver skin differently from the spent coffee ground seems questionable. The point that silver skin was part of the coffee powder and later of the spent coffee grounds for the major part of the use history should be taken into account when assessing novel food status.

## Figures and Tables

**Figure 1 molecules-27-06839-f001:**
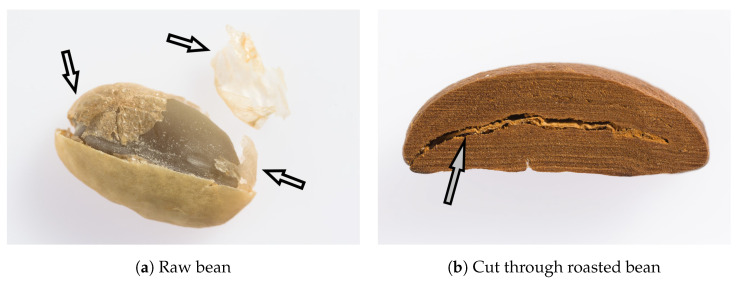
Macroscopic pictures of coffee beans (silver skin marked with arrows). (**a**) Raw coffee bean with silver skin exposed on upper half, silver skin partly detaching from bean. (**b**) Cut through roasted coffee bean showing the remaining silver skin between the two halves of the bean. Reproduced with permission from Fotodesign Luca Siermann, Wald-Michelbach, Germany.

**Figure 2 molecules-27-06839-f002:**
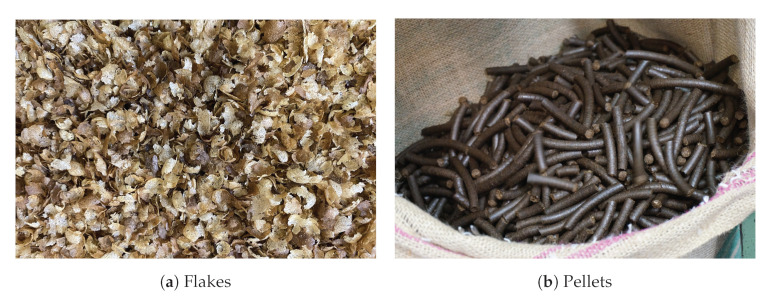
Processed silver skin. (**a**) Silver skin as flakes. (**b**) Silver skin pressed into pellets. (Photos by the authors).

**Figure 3 molecules-27-06839-f003:**
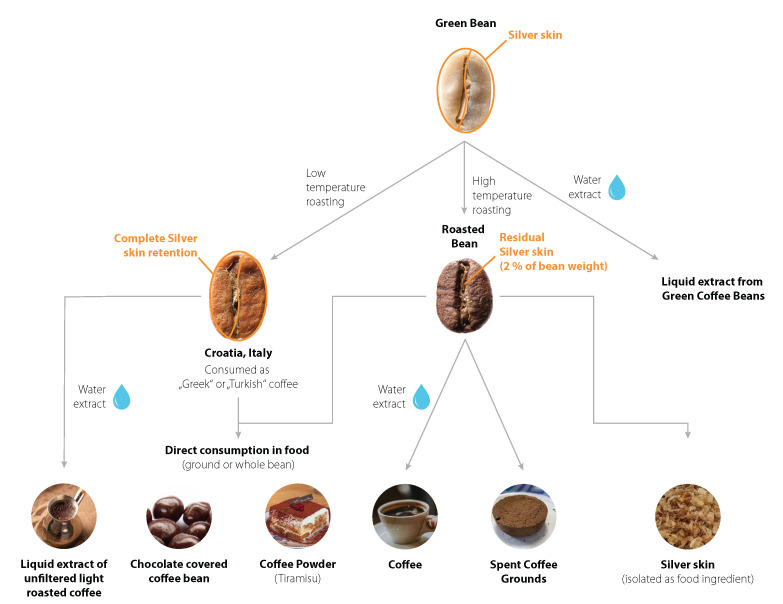
Various direct and indirect forms of human consumption of silver skin from coffee (the content of silver skin shown in yellow). Starting from the unroasted green coffee bean, roasting and brewing lead to the various coffee products. While all other shown products containing silver skin are not novel, isolated silver skin was considered as a novel food.

**Table 1 molecules-27-06839-t001:** Macro nutrients of coffee silver skin.

Nutrient	Amount (%)	Reference
Fats ^a^	1.6–3.0	[3,27,28,29,30,31,32]
Protein	7.1–22.0	[3,29,31,33]
Carbohydrates	9.5–14.5 (70.2) ^b^	[31,34]
fiber (total)	34.7–68.5	[28,29,31,34,35]
– soluble	8.2–11.0	[3,29]
– insoluble	46.0–56.0	[3,29]
Moisture	2.0–7.0	[30,31,36]
Ashes	5.4–9.5	[3,29,30]

^a^ Mainly saturated fatty acids. ^b^ Please note the value in brackets for carbohydrates also includes dietary fiber.

**Table 2 molecules-27-06839-t002:** Plant secondary compounds of coffee silver skin, with possible effects [45,46].

Ingredient	Amount	Possible Effect	Reference
Caffeine	0.65–1%	stimulant, diuretic, antioxidant	[3,31]
Polyphenols, including:	1–6 mg GAE ${}^{1}$/g	antibacterial, enzyme inhibitory, antioxidant	[33,47,48]
Chlorogenic acids	1.1–6.8%	antioxidant, antidiabetic, antiinflammatory	[49]
Vannilic acid	0.088–0.147%	antioxidant, flavouring	[27]
Syringic acid	0.009–0.036%	antioxidant	[27]
Flavonoids, including:	0.18–2.35 mg RE ${}^{2}$/g	antioxidant	[46]
Rutin	0.001–0.005%	antioxidant	[27]
Kaempferol- 3-glucoside	0.003–0.007%	antioxidant	[27]
Melanoidins	17–23%	antioxidant, anticancer, anticholesterol	[47]

^1^ GAE: gallic acid equivalent, ^2^ RE: rutin equivalent. All values were measured in extracts and calculated back to dry matter except flavonoids and chlorogenic acids which are given as amount in the extract.

**Table 3 molecules-27-06839-t003:** Contaminants of coffee silver skin. If not stated otherwise the regulatory standard is a limit.

Impurity	Amount	Reference	Regulatory Standard
Lead	<1–2.63 mg/kg	[35,42,44]	0.2 mg/kg (wheat) [55]
Mercury	0.05 mg/kg	[42,44]	TWI 4 μg/kg bw [58]
Cadmium	0.07 mg/kg	[42,44]	0.1 mg/kg (wheat) [56]
5-Hydroxymethylfurfural	0.57 mg/kg	[36]	Threshold 1.5 μg/person/d [57]
Furfuryl alcohol	n.d.	[3]	Threshold 1.5 μg/person/d [57]
Polycyclic aromatic hydrocarbons	traces	[44]	1 μg/kg Benzo(a)pyrene in babyfood [59]
Furan	n.d.	[44]	Threshold 1.5 μg/person/d [57]
Methylfuran	n.d.	[44]	Threshold 1.5 μg/person/d [57]
Pesticides	n.d.	[44]	depending on individual substance
Acrylamide	11.42 μg/L (Extract)	[3,35,40,60]	Benchmark:
	<20–161 μg/kg		400 μg/g
	720 μg/kg		(coffee) [61]
Phytosterol oxidation products (POP)	2.1–8.8 mg/kg	[23]	safe intake: 0.64 mg/kg bw/day [62]
Ochratoxin A	<4 μg/kg	[63]	5 μg/kg (coffee) [59]
	18.7–34.4 μg/kg	[64]	
Aflatoxins	B1 < 0.20 ppb	[34]	2 μg/kg (wheat) [59]
	B2 < 0.06 ppb		Sum of all
	G1 < 0.20 ppb		4 μg/kg (wheat) [59]
	G2 < 0.06 ppb		

n.d.: not detected, TWI: tolerable weekly intake.

**Table 4 molecules-27-06839-t004:** Estimation of intake of coffee silver skin from various sources.

Source	Concentration [%]	Estimated Intake of Source [g/kg bw/Day]	Maximum Worst-Case Intake from Source [g/kg bw/Day]
Flat Bread	5	0.89	0.045
Cakes	7	0.83	0.058
Biscuits	3	0.60	0.018
Cookies	5	0.99	0.050
Yoghurt	6	2.99	0.179
Burger Patties	3	0.79	0.024
Total assumed worst-case intake	0.374 g/kg bw/day		

**Table 5 molecules-27-06839-t005:** Nutritional data for relevant components of silver skin.

Component	Amount	Evaluation
Macro nutrients		
Fats	1.6–3.0% [3,27,28,29,30,31,32]	Low fat food
Carbohydrates	9.5–14.5% [31,34]	Low carbohydrate food
Proteins	15.0–22.0% [3,29,31]	All essential amino acids except methionine [3], relevant protein source
Fiber	34.7–68.5% [28,29,31,34,35]	Fiber source for nutrition, reduces deficiency
Micro nutrients		
Vitamin E	4.17 mg/100 g [36,41]	No relevant part of reference dose [43]
Zinc	0.7–2.2 mg/100 g [29,35,36,44]	No relevant part of reference dose [73]
Potassium	5000 mg/100 g [29,36,44]	Relevant amount of daily dose, no concern [73]
Calcium	500–1000 mg/100 g [29,36,44]	Relevant amount of daily dose, reduces deficiency [73]
Magnesium	200–2000 mg/100 g [29,36,44]	Above average intake, no concern [73]
Iron	8–80 mg/100 g [29,36,44]	Exceeding reference dose, low absorption [73], no concern
Contaminants		
Aflatoxins	n.d. [34]	No concern [59]
POP	2.1–8.8 mg/kg [23]	No significant contribution to daily intake [62], no concern
Ochratoxin A	<4–34.4 μg/kg [63,64]	No concern if ≤5 μg/kg [59]
Acrylamide	< 20–720 μg/kg [3,35,40,60]	No concern for low content [61], limit would be useful
Mercury	0.05 mg/kg [42,44]	Inorganic, limited adsorption, no concern [58]
Cadmium	0.07 mg/kg [42,44]	Below limit for wheat, no concern [56]
Lead	≤0.36 mg/kg 2.63 mg/kg [42,44,44]	Low lead qualities available, content must be mitigated [42,55]
Caffeine	0.65–1% [3,31]	Not exceeding intake guidelines [50], no concern

n.d.: not detected.

**Table 6 molecules-27-06839-t006:** Summary of studies of roasted coffee silver skin in various cell lines.

Effect Assessed	Test System	Cell Line	Outcome	Source
Cytotoxicity	MTT assay	HepG2	not cytotoxic up to 10,000 μg/mL	[65]
Cytotoxicity	LDH assay	HaCaT and HFF-1	negative up to 1000 μg/mL	[76]
Cytotoxicity	MTT assay	SH-SY5Y	not cytotoxic for all extracts tested	[77]
DNA damage	Comet assay	HepG2	negative up to 1000 μg/mL	[65]
Antioxidant capacity	Hemolysis	Erythrocytes	protective against oxidative stress at 25 g/L	[36]
Reduction of intracellular reactive oxygen species	ORAC Assay	IEC-6	reduction at 0.004, 0.04, and 0.4 mg/mL	[71]
Modulation of insulin secretion	ELISA	pancreatic INS-1E	enhanced secretion	[78]
Uptake of sugars	*non standard assay*	Caco-2	reduced uptake	[79]
Obesity effects	*non standard assay*	3T3-L1 RAW264.7 Mϕ	reduction of obesity benchmarks	[80]

Abbreviations: 3-(4,5-dimethylthiazol-2-yl)-2,5-diphenyltetrazolium bromide (MTT), human hepatocellular carcinoma cells (HepG2), lactate dehydrogenase (LDH), human immortalized non-tumorigenic keratinocyte cell line (HaCaT), human foreskin fibroblasts (HFF-1), human neuroblastoma cell line (SH-SY5Y), Oxygen Radical Absorbance Capacity (ORAC), rat small intestine epithelial cells (IEC-6), enzyme-linked immunosorbent assay (ELISA), rat insulinoma cell line (INS-1E), human intestinal epithelial (Caco-2) cells, mouse preadipocytes (3T3-L1), murine-leukemic monocyte-macrophage cell line (RAW264.7 Mϕ)).

**Table 7 molecules-27-06839-t007:** Overview of chronic animal studies.

Effect Assessed	Animal Species	Duration (d)	Outcome	Source
Toxicity of aqueous extract	rat	28	no toxic effects a 1 g/kg bw per day	[82]
Toxicity of melanoidin rich extract fraction	rat	28	no toxicity at 1 g/kg bw per day, slight effect of high fiber content	[71]
Effect of silver skin extracts on muscle growth	mouse	29	bigger forelimb muscle, more grip strength	[83]
Hypolipidemic effect of silver skin	rat	56	reduced effect of high fat nutrition	[84]
Effect of silver skin extract on streptozotocin induced damage	rat	35	reduction of damage	[72]
Effect of extract on total cholesterol and triglyceride plasma levels	rat	45	reduction, inhibition of pancreas lipase	[85]

## Data Availability

No new data were created or analyzed in this study. Data sharing is not applicable to this article.

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
