# Peer review of "Toxicological Assessment of Roasted Coffee Silver Skin (Testa of Coffea sp.) as Novel Food Ingredient"

_molecules, 2022, doi:10.3390/molecules27206839_

Round 1

Reviewer 1 Report

The manuscript entitled "Toxicological Assessment of Roasted Coffee Silver Skin (Testa of Coffea sp.) as Novel Food Ingredient" is a good work by the authors. Though, the review article has interesting points, there is a need of significant revision prior to acceptance for publication. Hence, I reccommend a major revision of the article prior to consideration as per the following comments.

1. The abstract needs to be significantly improved by incoporating the significance of the coffee silver skin.

2. The information about the production and consumption of the coffee is not mentioned in the introduction. I recommend to include these details along with the statistics on the production and use of coffee and producing nations

3. Figure 3 is good, however, the writings needs to be increased in size to enable easy reading

4. As shown in table 1 and Table 5., it contains around 3% of fat; however, the nature of fat (saturated or unsaturated) is not listed. It needs a clarification

5. In table 2, authors mentioned about polyphenols and flavonoids; these are broad spectrum classification. I recommend to mention the individual polyphenols or flavonoids present in the food component based on the literature.

6. Table 5 lack literature support. Inlcude the references

7. I recommend to include the details of health benefits as a separate sub-heading after the toxicity part. It will enhance the value of the review article

8. There are several typographic and punctuation errors in the manuscript. Kindly make a thorough proof reading.

Author Response

The manuscript entitled "Toxicological Assessment of Roasted Coffee Silver Skin (Testa of Coffea sp.) as Novel Food Ingredient" is a good work by the authors. Though, the review article has interesting points, there is a need of significant revision prior to acceptance for publication. Hence, I recommend a major revision of the article prior to consideration as per the following comments.

  1. The abstract needs to be significantly improved by incoporating the significance of the coffee silver skin.

RESPONSE: We agree that the great potential of silver skin is at best rudimentary represented in the abstract. We tried to add more potential uses and to clarify some points. On the other hand, the allowed length of abstract is very limited and as the major part of this article deals with the toxicity of silver skin and its novel food status, we have to represent this in the abstract as well. Therefore, in our opinion limiting the proportion of silver skin uses and potentials is necessary.

  1. The information about the production and consumption of the coffee is not mentioned in the introduction. I recommend to include these details along with the statistics on the production and use of coffee and producing nations

RESPONSE: Agreed. Giving more information about production and consumption is helpful to understand the relevance of silver skin as novel food. The information is added to the article.

  1. Figure 3 is good, however, the writings needs to be increased in size to enable easy reading

RESPONSE: Figure replaced as suggested.

  1. As shown in table 1 and Table 5., it contains around 3% of fat; however, the nature of fat (saturated or unsaturated) is not listed. It needs a clarification

RESPONSE: The saturated nature of the fats as well as the main fatty acids are given in the text (107-110). We agree that the information is not well visible. A footnote is added to the table, stating that mainly saturated fatty acids are part of the fat, to make the information better accessible.

  1. In table 2, authors mentioned about polyphenols and flavonoids; these are broad spectrum classification. I recommend to mention the individual polyphenols or flavonoids present in the food component based on the literature.

RESPONSE: The main polyphenolic components, chlorogenic acids, are already given in the table. As there is a sheer mass of polyphenols and flavonoids in the silver skin (comparable to coffee) a detailed consideration of single components would go beyond the possible extent of this article. Regarding toxicology in terms of novel food application, the effect of the mixture is more important than the effect of single components and regarding possible uses the antioxidant properties are the most relevant effects, therefore the mentioned broad spectrum is reasonable. But we agree that examples give a deeper understanding of silver skin composition, therefore we added the two / three components with highest content each to the table. Taking a deeper look into polyphenols and flavonoids of silver skin, the toxicological relevance of individual components and the difference from coffee to coffee silver skin is definitely a worthwhile content for an individual article.

  1. Table 5 lack literature support. Inlcude the references

RESPONSE: While this is hard for the evaluation column, which sums up the following text, we added the information for the contaminant contents and for the evaluation as far as specific sources were used. Due to this decision, the format of the table is now less legible, but this seems acceptable.

  1. I recommend to include the details of health benefits as a separate sub-heading after the toxicity part. It will enhance the value of the review article

RESPONSE: The description of the health benefits would considerably be beyond the scope of the current work and require another review paper. Additionally, the benefits are typically based on in vitro evidence or animal bioassays, but human clinical or epidemiological evidence is typically lacking. For that reason, we have decided to delete the few instances where we mentioned health benefits, to avoid confusion of the reader with incomplete evidence.

  1. There are several typographic and punctuation errors in the manuscript. Kindly make a thorough proof reading.

RESPONSE: Thank you very much for this hint. We re-checked our text and removed several errors.

Reviewer 2 Report

Dear Editors,

The article entitled “Toxicological Assessment of Roasted Coffee Silver Skin (Testa of Coffea sp.) as Novel Food Ingredient” is interesting. It collects many information related to coffee silver skin composition and toxicological status in order to look the possibility to use the skin as food ingredient. However, I have some notes and questions that may be useful for improving the article.

1.       Please avoid long sentences to increase the readability (e.g., line 2-4).

2.       Line 18-23 : Do the authors expect that coffee silver skin will be used to replace wheat / flour? If so, please elaborate more in this paragraph

3.       Figure 1: I will be better if the authors add some arrows directly pointing the silver skin

4.       Line 42: The coffee leaves and coffee] cherry pulp (or cascara) can be used for various purposes and were approved as traditional foods from third countries.  Where is the country? What is the name of traditional food?

5.       Figure 2: If the authors expect that coffee silver skin will be used to replace wheat / flour, it is much better if the picture of powdered silver skin is added.

6.       Line 78-79: What is the difference between this review and the review of Klinger et al.? Please explain

7.       Line 83: The authors stated that this review is “a systematic review”. However, I don’t see that this review was develop using systematic method (e.g. PRISMA) and thus still included as narrative review. Please confirm.

8.       Section 3.1.4.: For all type of the contaminant, please discuss where do they possibly come from and how to mitigate.

9.       Table 3: Please add one more column indicating the standard of a regulatory body (s) related to the contaminant.

10.   Table 5: Please add the references for making the table

After solving above-mentioned issue, this article is recommended for publication in this journal.

Author Response

Dear Editors,

The article entitled “Toxicological Assessment of Roasted Coffee Silver Skin (Testa of Coffea sp.) as Novel Food Ingredient” is interesting. It collects many information related to coffee silver skin composition and toxicological status in order to look the possibility to use the skin as food ingredient. However, I have some notes and questions that may be useful for improving the article.

  1. Please avoid long sentences to increase the readability (e.g., line 2-4).

RESPONSE: We removed most of the long sentences. A very limited number cannot be replaced, as doing so would make the text more complicated instead of easier.

  1. Line 18-23 : Do the authors expect that coffee silver skin will be used to replace wheat / flour? If so, please elaborate more in this paragraph

      RESPONSE: Silver skin does not aim to replace flour in larger amounts. It aims to be a value adding (more fiber, proteins, less calories) for several foods. Recent studies used it mainly as partial replacement for flour but silver skin is not limited to this application. We have added the word “partial replacement” on all instances to avoid the impression that a full replacement might be possible.

  1. Figure 1: I will be better if the authors add some arrows directly pointing the silver skin

RESPONSE: Replaced as suggested.

  1. Line 42: The coffee leaves and coffee] cherry pulp (or cascara) can be used for various purposes and were approved as traditional foods from third countries.Where is the country? What is the name of traditional food?

RESPONSE: Giving more information about further coffee by-products is helpful to understand the relevance of silver skin as novel food. The information is added to the article.

  1. Figure 2: If the authors expect that coffee silver skin will be used to replace wheat / flour, it is much better if the picture of powdered silver skin is added.

RESPONSE: The pictures show the recently available forms of silver skin. Powdered silver skin is not the form available on the market. Given silver skin will be used commercially in powdered form a picture would be useful but this is not the case yet. Besides, the picture of a brown fine powder would not be much informative.

  1. Line 78-79: What is the difference between this review and the review of Klinger et al.? Please explain

RESPONSE: First, the number of publications since Klingel’s review is significantly increased and as Klingel et al. reviewed several coffee by-products, the single by-product was not reviewed in such depth. Second this paper aims to check if the available data is sufficient for a novel food application, identify gaps and how to close them if necessary. There was also a considerable expansion of the literature in the last years.

  1. Line 83: The authors stated that this review is “a systematic review”. However, I don’t see that this review was develop using systematic method (e.g. PRISMA) and thus still included as narrative review. Please confirm.

RESPONSE: The reviewer is correct that we did not strictly follow PRISMA criteria. Therefore, we removed the word “systematic” in the methodology section. Nevertheless, we believe that we have comprehensively covered all available literature.

  1. Section 3.1.4.: For all type of the contaminant, please discuss where do they possibly come from and how to mitigate.

RESPONSE: This was already done in parts but definitely lacking completeness. The information is added to the article.

  1. Table 3: Please add one more column indicating the standard of a regulatory body (s) related to the contaminant.

RESPONSE: This really improves the table. Done as suggested.

  1. Table 5: Please add the references for making the table

RESPONSE: While this is hard for the evaluation column, which sums up the following text, we added the information for the contaminant contents and for the evaluation as far as specific sources were used. Due to this decision, the format of the table is less legible, but this seems acceptable.

After solving above-mentioned issue, this article is recommended for publication in this journal.

RESPONSE: Thank you for your evaluation.

Round 2

Reviewer 1 Report

There are no more comments.